Keraphyton gen. nov., a new Late Devonian fern-like plant from Australia

Champreux Antoine antoine.champreux@gmail.com 1 2
Meyer-Berthaud Brigitte 1
Decombeix Anne-Laure 1
1 AMAP, Université de Montpellier, CIRAD, CNRS, INRA, IRD , Montpellier , France
2 ARC Centre of Excellence for Australian Biodiversity and Heritage, Global Ecology, College of Science and Engineering, Flinders University , Adelaide , SA , Australia
Wilf Peter
Electronic publication date: 2020 Jun 16
Publication date: 2020
Volume: 8
Electronic Location ID: e9321
Received 2019 Dec 4; Accepted 2020 May 18
Copyright: ©2020 Champreux et al.
Copyright year: 2020
Copyright holder: Champreux et al.
License: This is an open access article distributed under the terms of the Creative Commons Attribution License, which permits unrestricted use, distribution, reproduction and adaptation in any medium and for any purpose provided that it is properly attributed. For attribution, the original author(s), title, publication source (PeerJ) and either DOI or URL of the article must be cited.
License URL: https://creativecommons.org/licenses/by/4.0/

Keywords: Palaeobotany, Palaeozoic, Early ferns, Gondwana, Australia, Plant anatomy, Iridopteridales

Funding: LabEx CeMEB (Mediterranean Center for Environment and Biodiversity Laboratory of Excellence) MARCON This work was supported by the LabEx CeMEB (Mediterranean Center for Environment and Biodiversity Laboratory of Excellence) for the Exploratory project MARCON (MARine and CONtinental ecosystems survival strategies in relation to global changes at the D-C boundary). The funders had no role in study design, data collection and analysis, decision to publish, or preparation of the manuscript.

==============================
The first plants related to the ferns are represented by several extinct groups that emerged during the Devonian. Among them, the iridopterids are closely allied to the sphenopsids, a group represented today by the genus Equisetum. They have been documented in Middle to early Late Devonian deposits of Laurussia and the Kazakhstan plate. Their Gondwanan record is poor, with occurrences limited to Venezuela and Morocco. Here we describe a new genus from a late Late Devonian locality of New South Wales. It is represented by a single anatomically preserved large stem characterized by a star-shaped vascular system with protoxylem strands located at rib tips, and by a lack of secondary tissues. Within the first fern-like plants, this stem shares the largest number of characters with iridopterid axes but differs by the pattern of its vascular system. Keraphyton mawsoniae gen. et sp. nov. adds a new record of early fern-like plants in eastern Gondwana. It provides new insights into the anatomical diversity within this key group of plants and supports the distinctiveness of the Australian flora in the latest Devonian.

Introduction

The Devonian is a time of major interest for understanding the origination and early phases of evolution of two major groups of plants, the ferns and the seed plants. Many analyses have focused on the early seed plants and their unique mode of reproduction, and a reasonable level of understanding of their patterns of diversification has been reached (Taylor, Taylor & Krings, 2009; Meyer-Berthaud, Gerrienne & Prestianni, 2018 and references herein).

This is not the case for the early representatives of the ferns and their allies, i.e., the sphenopsids, cladoxylopsids, and iridopterids, which are not well circumscribed in the fossil record and whose phylogenetic relationships are not fully understood (Cordi & Stein, 2005; Meyer-Berthaud, Soria & Young, 2007; Rothwell & Nixon, 2006; Taylor, Taylor & Krings, 2009; Xue, Hao & Basinger, 2010). Beck & Stein (1993) suggested that the extinct cladoxylopsids and iridopterids that thrived during the Middle and Late Devonian, together with the sphenopsids that are represented today by the Equisetales, may have been part of a natural group united by the possession of “permanent protoxylem” strands. Later, Kenrick & Crane (1997) recognized the permanent protoxylem character as the synapomorphy of a more extended clade, the Moniliformopses or sphenopsid-fern clade, which also included the Filicopsida. The relevance of the Moniliformopses has been heavily discussed. Today, all recent large-scale phylogenomic analyses resolve the ferns as monophyletic with the Equisetales as sister to the rest of the group (Pryer et al., 2001; Pteridophyte Phylogeny Group I, 2016; Qi et al., 2018; Shen et al., 2018). On the other hand, the discovery of new Devonian fossils affiliated with the sphenopsids and the iridopterids suggests that a close phylogenetic relationship between these taxa remains a plausible option (Xue, Hao & Basinger, 2010).

The Iridopteridales were identified by Stein (1982) as plants characterized by a unique set of derived features including a whorled organotaxis with ultimate appendages and branches potentially borne at the same nodes, the lack of secondary tissues, and an actinostele, i.e., a star-shaped column of primary vascular tissues. Six genera were referred to the Iridopteridales by Berry & Stein (2000). Three were based on anatomically preserved specimens, Asteropteris Dawson, Arachnoxylon Read (Stein, Wight & Beck), and Iridopteris Arnold (Stein) (Dawson, 1881; Stein, 1982; Stein, Wight & Beck, 1983). Two were represented by specimens preserved both as compressions and in anatomy, Ibyka Skog & Banks and Compsocradus Berry & Stein (Skog & Banks, 1973; Berry & Stein, 2000). The compression-based genus Anapaulia Berry & Edwards, which resembles Ibyka, was also included in the Iridopteridales (Berry & Edwards, 1996). The genus Rotoxylon Cordi & Stein was later added to this order despite its lack of an actinostelic vascular system (Cordi & Stein, 2005).

In this article, we describe a new iridopteridalean plant genus of Late Devonian age from the locality of Barraba in New South Wales, one of the rare localities of this age in eastern Australia to have provided anatomically preserved plant fossils of excellent quality (Chambers & Regan, 1986; Meyer-Berthaud, Soria & Young, 2007; Evreïnoff et al., 2017). We further discuss the significance of this discovery in relation to the stratigraphical and palaeogeographical occurrences of the Iridopteridales.

Materials & Methods

The present article is based on a single specimen collected at Barraba, in the New England region of New South Wales, Australia. It is part of a collection of anatomically preserved specimens discovered by Mr. John Irving, an amateur geologist, on the left bank of the Manilla River at Connors Creek crossing. The fossils occur in the Mandowa Mudstone, a formation of the Tamworth Belt consisting of a thick and monotonous sequence of dark laminated mudstones interbedded with thin layers of creamy siltstones and sandstones (Vickery, Brown & Percival, 2010). The Mandowa Mudstone sediments are marine and correspond to a distal shelf and continental slope environment. A late Famennian age (Late Devonian) has been assigned to the plant beds based on two lines of evidence (Wright, 1988; Vickery, Brown & Percival, 2010): (i) the large number of remains of Leptophloeum australe they contain. This lycopsid is believed to be restricted to the Late Devonian; (ii) the occurrence of the Tournaisian conodont Siphonodella quadruplicata in nearby beds of the same formation which are higher stratigraphically. Two trilobites associated with the plants are consistent with a late Famennian age (Wright, 1988).

The 90 mm long specimen representing Keraphyton is housed in the Geological Survey of New South Wales and referenced under number MMF44986 (Mining Museum Fossil Collection number 44986, Fig. 1A). Its mineral composition was analyzed with a FEI Quanta 200 FEG TEM/SEM from the MEA (Analytical Electron Microscopy) platform at University of Montpellier. Calcium phosphate is the most abundant mineral in the specimen that also contains pyrite in places. Sulfur and iron are abundant components of the surrounding matrix.

To avoid fragmentation, we embedded the fossil into a low viscosity two components epoxy resin (Araldite® AY103/HY991) before sectioning it into five blocks (A to E). We prepared eight transverse thin sections, one on each cutting face (AS, BI, BS, CI, CS, DI, DS and EI), and two longitudinal thin sections in block A (AL1 and AL2). The specimen is not well preserved in blocks C to E and the following description is based mainly on sections made in blocks A and B.

Figure 1 Keraphyton mawsoniae gen. et sp. n., holotype.

General features. (A) Specimen before preparation. (B) General view of stem showing the 4 rib systems (Ia, Ib, IIa and IIb; see also Fig. 2 for interpretation). (C) Central segment and four fundamental ribs. (D) Rib system Ib showing a short branch dividing into two equal ultimate ribs at right and a long branch producing at least three long ultimate ribs at left. (E) Long branch of rib system IIa producing short ultimate ribs. (F) Short branch of rib system IIa dividing into two ultimate ribs. (G) Long branch of rib system IIb producing short ultimate ribs. (H) Long branch of rib system Ia producing long, but broken, ultimate ribs. cs: central segment; fr: fundamental rib, ic: inner cortex, oc: outer cortex, Lb: long branch, sb: short branch. Yellow arrowheads indicate ultimate ribs. All views except (A) from transverse section MMF44986 BI1. (D–H) are all oriented with the cortex of the axis towards the top of the photo. All scale bars except (B) 500 μm, (B) 2 mm.

Thin sections were observed with an Olympus SZX12 stereomicroscope and an Olympus BX51 microscope. We used the Archimed and Olympus cellSens imaging software packages for acquiring digital photographs and measuring cells. We used the AutoStitch image stitching software for reconstructing fine resolution images from multiple photos (Brown & Lowe, 2007). The drawing in Fig. 2A was made with a camera lucida fixed on the stereomicroscope. We described the specimen with the terminology of Stein (1982), which is usually employed to describe Devonian actinostelic axes.

The electronic version of this article in Portable Document Format (PDF) will represent a published work according to the International Code of Nomenclature for algae, fungi, and plants (ICN), and hence the new names contained in the electronic version are effectively published under that Code from the electronic edition alone. In addition, new names contained in this work which have been issued with identifiers by IPNI will eventually be made available to the Global Names Index. The IPNI LSIDs can be resolved and the associated information viewed through any standard web browser by appending the LSID contained in this publication to the prefix “http://ipni.org/”. The online version of this work is archived and available from the following digital repositories: PeerJ, PubMed Central, and CLOCKSS.

Systematic palaeobotany

Class CLADOXYLOPSIDA Pichi-Sermolli (1959)	
Order IRIDOPTERIDALES Stein (1982)	
Family INCERTAE SEDIS	
Genus KERAPHYTON gen. nov.	

Type species. Keraphyton mawsoniae Champreux, Meyer-Berthaud, Decombeix sp. nov.

Derivation of name. From κερας (kéras), the ancient greek for horn, and φυτòυ (phyto), plant. The generic name, Keraphyton, refers to the horn-shaped outline of the primary xylem ribs of the stem in transverse section.

Diagnosis. As for type species, see below:

Keraphyton mawsoniae sp. nov.	

Diagnosis. Stem up to 20 mm in diameter, with primary tissues only. Vascular system actinostelic, consisting of four fundamental ribs united to a central segment. Fundamental ribs branching unequally, one branch dividing into two equal ultimate ribs, the other dividing in more ultimate ribs of distally decreasing dimensions. Protoxylem strands exarch to mesarch, at tip of ultimate ribs. Metaxylem tracheids from 20 to 140 µm in diameter, the smallest ones arranged in 1-2 layers along the lateral edges of the ribs. Tracheid walls showing scalariform to multiseriate bordered pit pairs with elliptical apertures. Endodermis-like cells consisting of rectangular cells up to 200 µm high and 120 µm periclinally. Inner cortical cells up to 160 µm in diameter, thin-walled, polygonal to circular in transverse section, with transverse to oblique endwalls in longitudinal section. Outer cortex homogeneous, with cells becoming thicker-walled and narrower towards periphery.

Holotype. Specimen MMF44986, Palaeontological reference collections, Geological Survey of New South Wales, Australia.

Type locality. Barraba, New England, New South Wales, Australia; left bank of the Manilla River, upstream from the Connors Creek crossing.

Stratigraphy. Mandowa Mudstone Formation, Parry Group, Tamworth Belt. Famennian, Upper Devonian.

Derivation of name. In honor of Prof. Ruth Mawson, distinguished Australian palaeontologist who was interested in all aspects of fossil life and was a delightful leader of palaeobotany-palaeontology field trips to Devonian localities of north-eastern Australia.

Description

The Barraba specimen is a straight, 90 mm long and 20 × 10 mm wide portion of stem (Fig. 1A). It has been compressed and is slightly fluted throughout its length. There is no external evidence of any branch or lateral organ.

The stem contains only primary tissues and is actinostelic (Fig. 1B). Some parts of the xylem at rib tips, as well as the phloem and associated tissues, are poorly preserved and replaced by a lacuna containing dark material. The inner cortex is made of thin-walled cells that merge progressively to the thicker-walled elements of the outer cortex. All cortical cells are elongated in longitudinal section. The outermost cortical layers and epidermis are missing. One side of the specimen at least is surrounded by a dark amorphous material that may have resulted from the degradation of the outermost cell layers. No vascular trace is observed at any level in this specimen.

Figure 2 Keraphyton mawsoniae gen. et sp. n., holotype.

(A) Drawing of specimen in transverse section; xylem in dark grey, cortex in light grey. (B) Schematic outline of xylem. (C) Schematic drawing of xylem structure.

Stele

The primary xylem is deeply ribbed in transverse section (Figs. 1B and 2A). Using Stein’s (1982) nomenclature, it consists of four fundamental ribs united by a central segment about 2.5 mm long (Figs. 1C and 2B). The fundamental ribs bifurcate unequally. The two resulting branches have different length (Figs. 1B, 2B). Long branches are more extended and divide more times than short branches. Divisions in long branches produce three to four ultimate ribs, the distalmost ones being the shortest (Figs. 1E, 1G and 1H). Short branches divide only once and produce two ultimate ribs that are approximately equal in dimensions (Figs. 1D and 1F). The four systems of at least six ultimate ribs (Ia, Ib, IIa, IIc in Fig. 2C) produced by the fundamental ribs are organized in two opposite pairs (pairs I and II). The two systems of a pair (e.g., Ia and Ib) exhibit a 180° rotational symmetry. Pair I rib systems (Ia and Ib) differ from pair II (IIa and IIb) by (i) longer subtending fundamental ribs, (ii) more extended long branches, (iii) longer ultimate units produced by long branches (Figs. 2B and 2C). The fundamental ribs range from five to ten cells and are 400 to 500 µm wide (Fig. 1C). Those subtending pair I rib systems are about 3.5 mm long, those subtending pair II rib systems are shorter; the reconstructed length of the latter, after adding their broken parts, is about 2 mm (Figs. 1B and 2C).

Metaxylem tracheids are polygonal with rounded corners in transverse section (Figs. 3A–3D). They measure up to 140 µm in diameter. The smallest tracheids are 20–35 µm wide and arranged in 1–2 layers along the lateral edges of the ribs (Fig. 3C). There is no protoxylem strand in the central segment nor in the fundamental ribs (Figs. 3A and 3C). Protoxylem strands are exarch to mesarch at the tips of the ultimate ribs (Figs. 3B and 3D). They are not associated with any thin-walled cell nor lacuna. In longitudinal section, metaxylem tracheid walls show elongated bordered pit pairs which are either uniseriate (i.e., scalariform) or multiseriate (Figs. 4A and 4B). Apertures are elliptical.

Figure 3 Keraphyton mawsoniae gen. et sp. n., holotype.

Detailed anatomy in transverse section. (A) Central segment showing wide tracheids in the median plane, even in the most compressed zones. (B) Tip of ultimate rib showing small tracheids interpreted as elements of exarch to mesarch protoxylem strand (arrowhead). (C) Fundamental rib showing small tracheids on lateral edges; well-preserved inner cortical cells at top. (D) Tip of ultimate rib showing small tracheids interpreted as elements of exarch to mesarch protoxylem strand (arrowhead). (E) Endodermis-type cells bordering highly compressed inner cortical cells; displaced xylem tracheids at left. (F) Endodermis-type cells. (G) Section through cortex showing large, thin-walled inner cortical cells at right and small, thick-walled outer cortical cells at left. Transition gradual between inner and outer cortex. (H) Lacuna around ultimate rib tip showing presumed remains of phloem tissue; endodermis-type cells at top. cs: central segment, en: endodermis-type cells, fr: fundamental rib, La: lacuna, ph: phloem tissue, ic: inner cortex, oc: outer cortex. All views except (D) from transverse section MMF44986 BI1, (D) from transverse section MMF44986 CI1. All scale bars 200 μm.

Figure 4 Keraphyton mawsoniae gen. et sp. n., holotype.

Detailed anatomy in longitudinal section. (A) Section showing from left to right xylem tracheids, lacuna and endodermis-type cells, highly compressed inner cortical elements and elongated thick-walled outer cortical cells. (B) Xylem tracheids showing elongated bordered pits with elliptical apertures. (C) Inner cortical cells with transverse end walls (D) Outer cortical cells with progressively smaller diameters and thicker walls to the right. Note small holes in the cell walls. en: endodermis-type cells, La: lacuna, ic: inner cortex, oc: outer cortex, xy: xylem. All views from transverse section MMF44986 ASl1. All scale bars 200 μm.

The framboidal pyrite-filled lacuna surrounding the xylem is about 200 µm wide. It may attain a width of 1,000 µm around rib tips and contain patches of preserved cells in these areas. We tentatively interpret groups of small (14–38 µm wide) thin-walled elements as fragments of phloem tissue (Fig. 3H). Pyrite framboids are often documented in permineralized plants and usually develop before the mineralization as a product of the degradation of organic matter (Garcia-Guinea, Martínez-Frías & Harffy, 1998). The presence of framboidal pyrite in the phloem region indicates the preferential degradation of these tissues.

Cortex

On its inner side, the cortex is bounded by one to two layers of cells with a rectangular shape in transverse section (Figs. 3E, 3F and 3H). They may represent an endodermis. These cells are 50 to 120 µm in the periclinal dimension and 30–75 µm radially. In longitudinal section they show transverse end-walls and their height ranges from 80 to 200 µm (Fig. 4A).

The inner cortical cells are laterally compressed (Fig. 3E). When their original shape and size are preserved, these thin-walled elements appear polygonal to circular in transverse section (Figs. 3C and 3G), and elongated with transverse to oblique end-walls in longitudinal section (Fig. 4C). Some are filled with pyrite. They measure 35–160 µm in diameter and are 90–400 µm long.

The outer cortex is homogeneous. Outer cortical cells show progressively smaller diameters and thicker walls towards the stem periphery (Fig. 3G). Diameter decreases from 140 to 25 µm and wall thickness increases from about 7 to 18 µm outwardly. In longitudinal section, the shape and length of the outer cortical cells do not differ much from those of the inner cortical cells (Fig. 4D). Numerous small oval to circular holes on the walls may represent pits or early signs of degradation.

Vascular traces and organotaxis

There is no evidence of vascular trace emission in the best-preserved parts of the specimen but the structure of the primary xylem as illustrated in Figs. 1B and 2 provides some information on the organotaxis. The ultimate ribs produced by the two long branches of pair I rib systems are comparable in size and they are more elongated than those of pair II (compare Figs. 1D and 1H with Figs. 1E, 1G; Figs. 2B, 2C). This pattern is potentially linked to the emission of incipient traces by rib systems Ia and Ib. Rib systems IIa and IIb correspond to a different developmental stage regarding the production of traces to lateral organs.

How exactly each rib system contributed to the vascular supplies of lateral organs and how many types of lateral organs were borne at each node (i.e., did short and long branches contributed to different types of lateral organs) are highly conjectural. Different hypotheses are proposed in Fig. 5. All assume that each ultimate rib produces a vascular trace. In Fig. 5A, each ultimate rib produces a trace that innervates one lateral organ. In Fig. 5B, two different types of lateral organs are produced per node, a small type showing the two traces generated by the short branches, a large type showing the traces generated by the long branches. In Fig. 5C, the traces produced by each rib system run into a single lateral organ. In all cases, the arrangement of the lateral organs, whether borne singly or in groups, is opposite decussate.

Figure 5 Keraphyton mawsoniae gen. et sp. n., holotype.

Hypotheses about vascular trace production and lateral organ arrangement in stems. (A) Each trace generated by an ultimate rib runs into one lateral organ; all lateral organs similar. (B) Paired traces generated by ultimate ribs on short branches run into a small type of lateral organ, group of traces generated by ultimate ribs on long branches run into a large type of lateral organ. (C) All traces generated by the ultimate ribs of a rib system run into a single large lateral organ.

Discussion

Stem or root?

The deeply ribbed vascular system of Keraphyton, the fact that it possesses an endodermis and that its protoxylem strands may be exarch suggest that it represents a root. These three traits, however, are not restricted to roots. Actinostelic vascular systems are common in stems of Devonian age affiliated to the lignophytes (e.g., the Stenokoleales, aneurophytalean progymnospems and early seed plants) and to fern-like plants such as the Iridopteridales (Taylor, Taylor & Krings, 2009; Momont, Gerrienne & Prestianni, 2016). Among extant plants, an endodermis is commonly found in stems, whether rhizomatous or aerial, of Equisetum, the Psilotales, Ophioglossales, Marattiales, Osmundales, and Polypodiales (Bell & Hemsley, 2000). In the Devonian an exarch maturation of the primary xylem occurs in stems affiliated to the Cladoxylopsida and the Sphenophyllales (Table 1). In addition, the pattern of vascular trace production anticipated for laterals in the Keraphyton mawsoniae type-specimen is that of a stem. If it were a root, a proliferation of cells at the endoderm level would be visible at the site of lateral root production (Motte & Beeckman, 2019). We therefore interpret the available specimen of Keraphyton mawsoniae as a fragment of stem about to produce lateral organs.

Affinities

There is no information on the lateral organs, their nature, size and arrangement in Keraphyton. Nevertheless, this stem provides sufficient features to demonstrate its uniqueness and its affiliation to a new genus within the “permanent protoxylem” group proposed by Beck & Stein (1993).

Based on developmental hypotheses involving the role of hormones in the differentiation of the primary vascular tissues, Beck & Stein (1993) distinguished two main groups among the numerous plants of Devonian and early Carboniferous age characterized by an actinostelic vascular system. In the “radiate protoxylem” group that includes some basal euphyllophytes, such as the Stenokoleales, the aneurophytalean progymnosperms, and some earliest seed plants, protoxylem strands occur along the midplanes of the xylem ribs and they all derive by branching from a single permanent strand located centrally. In the “permanent protoxylem” group, protoxylem strands are exclusively peripheral, a pattern that may reflect a hormonal prominance of the lateral organ meristems compared to the hormonal influence of the shoot apical meristem. With its protoxylem strands located only at rib tips and its largest tracheids occupying the median plane of the ribs, Keraphyton is clearly a member of the “permanent protoxylem” group that encompasses the pseudosporochnalean and non-pseudosporochnalean cladoxylopsids, the Sphenophyllales and the Iridopteridales.

Table 1 Comparison of main stem characters in the “permanent protoxylem” group.

	Cladoxylopsids	Sphenophyllales	Iridopteridales	Dixopodoxylon	Keraphyton	
Organotaxis	Helical & whorled	Whorled	Whorled	Unknown	Helical unlikely	
Organ types at nodes	1 type: branch	2 types: branch & appendage	2 types: branch & appendage	Unknown	Possibly 2 types	
Fundamental stelar ribs	Numerous	3, rarely 4	3–5, numerous in some taxa	7	4	
Connection of fundamental ribs	Not permanent	Permanent	Permanent (except Rotoxylon)	Permanent	Permanent	
External division of fundamental ribs	In some taxa	Absent	Present	Present	Present	
Division type of fundamental ribs	Equal	Inapplicable	Equal	Equal	Unequal	
Protoxylem	Mesarch; exarch at some levels in Polyxylon	Exarch & mesarch	Mesarch	Exarch to mesarch	Exarch to mesarch	
Parenchyma or lacuna with protoxylem	In some taxa	In some taxa	Yes	No	No	
Secondary xylem	In some taxa	Common	Rare	Absent	Absent	

The Pseudosporochnales, and the non-pseudosporochnalean cladoxylopsids such as Cladoxylon Unger, Polyxylon (Read & Campbell) Chambers & Regan, and Pietzschia Gothan, are characterized by a dissected stele composed of a much higher number of xylem ribs than Keraphyton (Unger, 1856; Gothan, 1927; Chambers & Regan, 1986; Soria & Meyer-Berthaud, 2005; Taylor, Taylor & Krings, 2009). Some ribs may temporarily connect internally, forming U-, V- or W-shaped patterns in transverse section but, unlike that of Keraphyton, the stele is never stellate with all ribs permanently connected at their inner extremity.

Five sphenophyllalean genera of Late Devonian age from the Hubei and Zhejiang provinces of China (South China plate) and Belgium (Laurussia) are represented by specimens that are partly or entirely preserved in anatomy (Ma, Liao & Wang, 2009; Terreaux de Félice, Decombeix & Galtier, 2019). These are Sphenophyllum Brongniart, Eviostachya Stockmans, Hamatophyton (Gu & Zhi) Wang, Hao, Tian & Xue, Rotafolia (Wang, Hao & Wang) Wang, Hao, Wang & Xue, and probably Pleurorhizoxylon Zhang, Berry, Wang, Xue & Liu (Brongniart, 1828; Stockmans, 1948; Wang et al., 2006a; Wang et al., 2006b; Zhang et al., 2018; Terreaux de Félice, Decombeix & Galtier, 2019). Sphenophyllalean axes share two characters with Keraphyton, a ribbed protostele and a primary xylem maturation that may be exarch. They differ from Keraphyton by a simpler stele that shows only three undivided short lobes. In the genus Rotafolia, the stele may occasionally be tetralobate (Wang et al., 2006b), but lobes are short and undivided, unlike the long, thin and multi-divided ribs of Keraphyton. The actinostele in Keraphyton is built on a four-arm pattern with much more extended ribs. Sphenophyllalean axes do not exceed 15 mm in diameter. Despite their small size, and unlike the Keraphyton type-specimen, they often show a well-developed secondary xylem.

Within the “permanent protoxylem” group, Keraphyton shares the largest number of characters with the Iridopteridales (Stein, 1982; Berry & Stein, 2000). Keraphyton contains only primary tissues. Since Stein’s (1982) diagnosis mentioning this character, species such as Arachnoxylon minor Stein, Wight & Beck and Rotoxylon dawsonii Cordi & Stein, which occasionally show a secondary-type of xylem, have been recognized as iridopteridalean (Stein, Wight & Beck, 1983; Cordi & Stein, 2005). Despite its relatively large diameter, the Keraphyton stem is devoid of any tissue of this type and is more consistent with the original concept of the order as defined by Stein (1982). In addition, the primary vascular system of Keraphyton consists of a deeply ribbed column of vascular tissue showing radially oriented xylem ribs that are united centrally. These characters also match the original definition of the Iridopteridales.

Iridopterids are characterized by a whorled organotaxis and the presence of both branches and ultimate appendages at nodes. There is no lateral organ attached to the type-specimen of Keraphyton, and its anatomical preservation does not allow an accurate analysis of how vascular traces were produced. Several possibilities of trace emission and organotaxis can be suggested. A helical organotaxis, however, is unlikely given the shape of the Keraphyton stele and the paired organization of its rib systems. This pattern rather suggests that there were at least two lateral organs arranged oppositely at each node. Moreover, if the short and long branches of each rib system contributed to the vascularization of two different types of organs, then four lateral organs, at least, may have been produced at each node in Keraphyton. Therefore, the arrangement of the lateral organs in the new genus may have been close to that of the Iridopteridales.

Keraphyton is characterized by specific features that set it aside from all other iridopteridalean taxa known to date and justify its assignment to a new genus (Fig. 6). Apart from Rotoxylon that shows undivided ribs (Fig. 6Q), fundamental ribs in the Iridopteridales divide equally forming two or three branches of similar dimensions. This is the case in Iridopteris (Figs. 6A and 6B), Asteropteris (Fig. 6I), Arachnoxylon (Figs. 6C, 6D, 6J, 6K and 6L), Compsocradus (Fig. 6H), Asteropteris (Fig. 6I), Ibyka (Fig. 6M) (Bertrand, 1913; Skog & Banks, 1973; Stein, 1982; Stein, Wight & Beck, 1983; Berry & Stein, 2000). This is also the case in Denglongia (Xue & Hao) Xue, Hao & Basinger (Fig. 6P), an actinostelic genus that has not been assigned to any specific order but whose possible iridopterid affinities have been investigated (Xue, Hao & Basinger, 2010). In contrast, fundamental ribs of Keraphyton divide asymmetrically, resulting in the short and long branches reported above. Protoxylem strands in Keraphyton are less conspicuous than in the other iridopteridalean taxa, they look exarch and are not associated with lacunae. Finally, the maximum diameter of metaxylem tracheids in Keraphyton are up to 140 µm wide, a diameter that largely exceeds that of the other iridopteridalean species which rarely reaches 100 µm.

Figure 6 Vascular system of Iridopteridales and allies in transverse section.

(A)–(B) Iridopteris eriensis. (C)–(D) Arachnoxylon minor. (E)–(G) Metacladophyton tetraxylum. (H) Compsocradus laevigatus. (I) Asteropteris noveboracensis. (J)–(L) Arachnoxylon kopfii. (M) Ibyka amphikoma. (N) Metacladophyton ziguinum. (O) Keraphyton mawsoniae. (P) Denglongia hubeiensis. (Q) Rotoxylon dawsonii. (R) Serripteris feistii. (S) Dixopodoxylon goense. Scale bar five mm. Secondary-type xylem in dark grey. (A)–(D), (I)–(M): adapted from Stein (1982); (E)–(G): adapted from Wang & Geng (1997); (H): adapted from Berry & Stein (2000); (N): adapted from Wang & Lin (2007); (P): adapted from Xue, Hao & Basinger (2010); (Q): adapted from Cordi & Stein (2005); (R): adapted from Rowe & Galtier (1989); (S) adapted from Fairon-Demaret (1969). [6P is republished with permission of University of Chicago Press, from “Anatomy of the Late Devonian Denglongia hubeiensis, with a discussion of the phylogeny of the Cladoxylopsida”; Xue J, Hao S, Basinger JF; 171, no. 1; 2010; permission conveyed through Copyright Clearance Center, Inc.].

The Keraphyton mawsoniae type-specimen is one of the largest iridopterid axes known to date. It is comparable in diameter to the type-specimens of Asteropteris noveboracensis and Rotoxylon dawsonii. Its primary vascular system, however, is markedly different (compare Fig. 6O with Figs. 6I and 6Q). Contrary to Keraphyton, the fundamental ribs of Rotoxylon are undivided, and a few ones only are centrally connected. The stele of Asteropteris shows no symmetry and the fundamental ribs, before division, are much shorter. The vascular system of Keraphyton differs also from that of the largest specimens of Metacladophyton Wang & Geng (Wang & Lin, 2007), a genus of Givetian to early Frasnian age represented by two species from Hubei. Metacladophyton is not included in the iridopterid-sphenophyllalean group recognized in Xue, Hao & Basinger’s (2010) cladistic analysis, but it has been proposed as a possible representative of the Iridopteridales by Berry & Stein (2000). In Metacladophyton, stelar ribs are not connected centrally (Figs. 6E, 6G and 6N). A connection of the stelar ribs has been observed in small axes only (Fig. 6F). The largest axes differ from Keraphyton not only by a different stelar configuration but also by the possession of a large amount of a secondary-type of xylem surrounding the stelar ribs (compare Fig. 6O with Figs. 6G and 6N).

The monospecific genus Dixopodoxylon Fairon-Demaret was erected for a small anatomically preserved specimen of Middle Devonian age from Belgium showing a deeply ribbed actinostele (Fairon-Demaret, 1969) (Fig. 6S). Dixopodoxylon goense has uncertain affinities but, if its type-specimen does not represent a root, its characters are consistent with those of the “permanent protoxylem” group of plants. Like Keraphyton, Dixopodoxylon is devoid of secondary xylem. Its primary xylem maturation is described as mesarch, but the protoxylem strands occur at rib tips, close to the xylem border. They are not associated with parenchyma or a lacuna as is Keraphyton. Dixopodoxylon, however, differs from Keraphyton by a larger number of fundamental ribs that divide equally rather than asymmetrically.

Table 1 summarizes the main points discussed in this section. In the “permanent protoxylem group”, Keraphyton shares numerous characters with stems affiliated to the Sphenophyllales and the Iridopteridales. Its highly divided actinostele and lack of secondary xylem makes it closer to the Iridopteridales. The main differences with taxa currently included in this order are the following: a higher number of divisions of the actinostele, the asymmetrical division of the fundamental ribs, and protoxylem strands that are not associated with either a lacuna or parenchyma. Would it be legitimate to create a new order based on these characters? We think that is it would be premature for the following reasons: (i) the number and symmetry of the divisions of the actinostele are not assessed in the diagnosis of the Iridopteridales. Adding Keraphyton in this order does not change its concept; (ii) the new order would be represented by a single specimen preserved on a short length that may not express the range of variation of its characters. We cannot exclude that, like several other taxa of the “permanent protoxylem” group (see Table 1), Keraphyton showed a lacuna or parenchyma associated with the protoxylem strands at some levels of its aerial system.

Palaeogeographical and stratigraphical considerations

The Iridopteridales, so far comprising the genera Asteropteris, Arachnoxylon, Iridopteris, Ibyka, Compsocradus, and Anapaulia range from the upper Eifelian (Middle Devonian) to the Frasnian (early Late Devonian). They are recorded from Laurussia (eastern USA and Spitzbergen), Gondwana (Venezuela and Morocco), and the Kazakhstan plate (Xinjiang, northwestern China) (Bertrand, 1913; Skog & Banks, 1973; Stein, 1981; Stein, 1982; Stein, Wight & Beck, 1983; Berry & Edwards, 1996; Berry & Stein, 2000; Ma, Liao & Wang, 2009; Fu et al., 2011). Despite its dissected stele with numerous unconnected stelar ribs, the addition of Rotoxylon (Cordi & Stein, 2005) to this list of iridopterids does not change the stratigraphical and palaeogeographical ranges of the order.

The genus of Frasnian age from Hubei, Denglongia, is characterized by an actinostelic vascular system which shows many similarities with that of the Iridopteridales (Xue, Hao & Basinger, 2010). The only anatomical difference lies in the possession of two protoxylem strands in the rib tips which are enlarged. This enlargement is suggestive of the initiation of a division of the rib, a common process in the iridopterids. Our opinion, therefore, is that this anatomical difference is minor and that iridopterids conforming to the original concept of the order may well have been present in South China during the Frasnian time interval.

The youngest genus showing some iridopterid characters is Serripteris Rowe & Galtier, represented by a single specimen of Tournaisian (Early Carboniferous) age from southern France (Rowe & Galtier, 1989). Serripteris has an actinostelic vascular system. Its stele has a very simple shape and shows four undivided ribs, each one with a single permanent protoxylem strand at its tip (Fig. 6R). Serripteris, however, differs from the Iridopteridales in its helical arrangement of the lateral branches and lack of ultimate appendages at nodes (Rowe & Galtier, 1989). At this state of knowledge, the iridopterid affinities of Serripteris are uncertain and a stratigraphical occurrence of the Iridopteridales in the Carboniferous speculative.

It is clear from these considerations that, up to now, genera conforming to the original description of the Iridopteridales (Stein, 1982) ranged in time from the Eifelian to the Frasnian. They occurred in Laurussia, the Kazakhstan plate, and probably the South China plate. Their Gondwanan record is poor and only comprises Compsocradus from Venezuela (Berry & Stein, 2000) and Anapaulia from both Venezuela and Morocco (Berry & Edwards, 1996; Prestianni et al., 2012). Keraphyton is the first possible representative of the Iridopteridales reported from Australia. Its occurrence in the Famennian beds of the Mandowa Mudstone at Barraba may indicate that the stratigraphical range of the Iridopteridales went beyond the Frasnian and encompassed, at least, the whole Late Devonian. Moreover, Keraphyton expands the palaeogeographical range of the Iridopterid-like plants to the easternmost part of north Gondwana.

At the Barraba locality Keraphyton is associated with numerous axes of Leptophloeum australe preserved as adpressions together with anatomically preserved specimens of the lycopsid Cymastrobus irvingii, the non-pseudosporochnalean cladoxylopsid Polyxylon australe, and wood fragments referable to the archaeopteridalean progymnosperm genus Callixylon (Chambers & Regan, 1986; Meyer-Berthaud, Soria & Young, 2007; Evreïnoff et al., 2017). Leptophloeum australe and Callixylon were cosmopolitan taxa in the Late Devonian but Cymastrobus, Polyxylon autrale, and now Keraphyton have not been recorded elsewhere, supporting the distinctiveness of at least part of the vegetation in eastern Australia at this time.

Conclusions

We describe a new genus of fern-like plants, Keraphyton gen. nov., from a Famennian locality of north-eastern New South Wales, Australia. It is represented by a large anatomically preserved stem showing iridopteridalean characters. This discovery shows that the Iridopteridales spanned the whole Late Devonian and occurred in the far east of Gondwana. After Polyxylon australe, a species from the same locality affiliated to the cladoxylopsids, this is the second report of an early diverging fern-like plant in the late Late Devonian of eastern Gondwana. Keraphyton mawsoniae and Polyxylon australe are unknown elsewhere, supporting the distinctiveness of at least part of the Australian flora during the Famennian.

We thank the Geological Survey of New South Wales and Dr. Yong Yi Zhen for the loan of the specimen; John Irving for information on the Barraba plant locality, Ian Percival and Earth Sciences members of the University of New England for advice on the geology of the area; Romain Blanchard who initiated this study; Frédéric Fernandez from the MEA (Analytical Electronic Microscopy) platform, University of Montpellier, for the TEM/SEM analyses. We also thank William Stein, two anonymous reviewers and the editor, Peter Wilf, for their comments and suggestions that greatly improved the original manuscript. AMAP (botAny and Modelling of Plant Architecture and vegetation) is a joint research unit which associates Montpellier University, French National Center for Scientific Research (Mixt Research Unit 5120), French International Cooperation Center for Agronomic Research and Development (Mixt Research Unit 51), French National Research Institute for Agriculture, Food and the Environment (Mixt Research Unit 931) and French Research Institute for Development (Research Unit 123).

Additional Information and Declarations

Competing Interests

Author Contributions

Data Availability

New Species Registration

The authors declare there are no competing interests.

Antoine Champreux, Brigitte Meyer-Berthaud and Anne-Laure Decombeix conceived and designed the experiments, performed the experiments, analyzed the data, prepared figures and/or tables, authored or reviewed drafts of the paper, and approved the final draft.

The following information was supplied regarding data availability:

The raw data consists of physical material, specifically a paleontological specimen loaned by the Geological Survey of New South Wales: Specimen MMF44986 (Mining Museum Fossil Collection number 44986), Palaeontological reference collections, available at the Geological Survey of New South Wales, Australia.

The following information was supplied regarding the registration of a newly described species:

Genus name: Keraphyton, Species name: Keraphyton mawsoniae.

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
