# Peer review of "Keraphyton gen. nov., a new Late Devonian fern-like plant from Australia"

_PeerJ, doi:10.7717/peerj.9321_

## Round 0.1 · original submission · Major Revisions

Thank you for submitting this interesting paleobotanical work to PeerJ. We have been fortunate to obtain reviews for you, even over the holiday break, from three dedicated colleagues with significant expertise in this group of plants. All agree that the paper reports an important fossil and that the illustrations are of high quality, while also giving numerous useful corrections and suggestions for improvement that are mostly minor in nature. I agree with all their suggestions.

Reviewer 3 poses alternative interpretations for the identity (somewhat echoed by reviewer 1) and interpretation (stem or root?) of the fossil that I agree should be carefully considered and addressed, perhaps requiring some significant modifications to the text.

I also agree with Reviewer 3 that the labeling of the illustrations needs significant improvement for clarity and visibility of the labels.

Your writing is very good for the most part, but I have made some suggestions for improvement in the markup pdf. The abstract needs a much stronger closing sentence. Please edit and check your revised text very carefully. There is no copy-editing at PeerJ, which helps keep the article charges reasonable, but detailed attention to the writing is therefore required.

Given the nature of some of the comments (esp. from Reviewer 3 ), and the large number of helpful suggestions from all three reviewers, I expect to send your revision for a second review to at least some of the original reviewers. Please include with your revision a detailed response letter explaining your accommodation or rebuttal of each reviewer suggestion.

I look forward to your revision.

·

Basic reporting

All of this is just fine. I have made a few minor suggestions linked to the text by line number. See General Comments section below.

Experimental design

Experimental design is sufficient although I would suggest addtion of preservation minerals in the Materials and Methods. All of this is explained in my review below.

Validity of the findings

I find the paper more-or-less as it stands perfectly acceptable. See General comments below for explanation

Additional comments

Review of:
Keraphyton gen. nov., a new Late Devonian plant genus of iridopterid affinities from New South Wales, Australia
Antoine Champreux, Brigitte Meyer-Berthaud, Anne-Laure Decombeix

I think this is an interesting report of important Devonian fossil material from Australia – a part of the ancient Earth from which we desperately need more information. In general, the description and analysis are well done, adequate for the purpose of typifying a new genus, and deserving of publication in PeerJ. I recommend the work more-or-less as the text already stands.
I list several comments by line number below mostly designed to clarify the presentation somewhat.
I offer to the authors the following comments that might be helpful:
1. Amplifying the presentation to include the nature of preservation is important, I think. See comment for text line 77.
2. I realize that all has been done that probably should be done given the limited nature of the specimen at hand. I certainly do not dispute the basic findings. However, Keraphyton represents a significant extension of the concept of Iridopteridales in time, morphology/anatomy, and geographic range that may or may not be useful. Only time will tell. In my opinion, Chinese, Moroccan and Australian material, from the late Devonian especially, seems to indicate the presence of groups at ordinal or class level that probably need to be established and defined in their own terms rather than appended to Middle Devonian high-level taxa from Europe or North America. Please note that this is not intended as a criticism. This is an exciting time in plant paleontology of these generally unknown regions, and I can think of no group in a better position to accomplish this task than the current authors.
Wm Stein
Line 21: I think you simply mean ‘pattern’ instead of ‘division pattern’ since a static pattern of the vascular system is observed and divisions of this system are in fact not observed.
Line 23: By ‘partly specific’ do you mean endemic or something else?
Line 49: The reference for Arachnoxylon perhaps suggests that Stein et al. were responsible for the generic name, when in fact this was Read’s original. We merely added a new species to an already standing genus in this paper. The same applies to Iridopteris; this name is Arnold’s. I think this is merely a matter of citing references (in parenthesis) apart from formal designation of taxonomy, i.e., Arachnoxylon Read or Iridopteris Arnold.
Line77: In this paragraph, no mention is made of the minerals that make up the specimen. For proper interpretation, I think adding this information would be very helpful. The specimen looks similar to late Devonian phosphatic permineralizations seen in the midcontinent North America. However, mention is made below of pyrite in the phloem region, I believe.
Line 78: In the previous paragraph reference is made to transverse and longitudinal ‘sections’. Were these all ‘thin sections’ as described here? If not, please specify where thin sections were made and where, perhaps, thick sections as opposed to blocks A-E still remain.
Line 165: I think I see what you refer to as protoxylem in transverse section, although it might be useful to label these cells in a figure as such. Were you able to verify identification of protoxylem in longitudinal section? Nothing is said about that in the text. I must say that this protoxylem, if that’s what it is, seems very different than the conspicuous permanent mesarch strands observed in the Iridopteridales.
Line 166: Here’s the mention of pyrite (=dark filled cells??) mentioned by me above. Although Fig. 3H shows some degree of compression, and therefore fairly densely packed with dark material, it appears from other views in Fig. 3, that the dark material comprises only one type of cell in the phloem region. In phosphatic permineralizations, we also observed dark-filled cells but, if memory serves, not pyrite. Dark-filled cells are also present in the pyrite & limonite permineralizations of Middle Devonian Iridopteridales. However, the dark stuff is, we guess, organic in origin.
196: I agree with your conjecture here. Thus, this might be reason to have concern about assignment with the Iridopteridales of the Middle Devonian at least. Might early ferns be an equally valid assignment?
Line 212: I might suggest saying something like ‘differential hormonal prominence of the shoot apex…’ or being more specific. Our original idea was that the central protoxylem in the Aneurophytales and others likely represented an image of the shoot apical meristem written by hormonal influence on developing provascular tissue much like, by analogy, light exposes film. The interpretation of Iridopteridales and others, by contrast, seemed to indicate less influence by the shoot apex and perhaps greater influence by the meristems of lateral branches or appendages. Your sentence as it stands might be interpreted to mean the opposite.
Line 216: I agree with this assessment, although some degree of intermediacy is observed within the Pseudosporochnales. Pitting in the xylem matches as well, but hypothesized organotaxy not, protoxylem organization not, and general form of the protostele probably not as well. So, it seems to me, the fossil presents us with a mixed message. How interesting!
Line 255: How so consistent? The Middle Devonian Iridopteridales all have whorled organotaxis. Keraphyton organotaxis, as hypothesized, seems to indicate a difference.
Line 269: The term “temporary” - a feature of time - is used to describe a static structural relationship. Perhaps it would be better to reword more precisely.
Line 286: I think these comparisons cover the situation in a reasonable way, but see my general comment 2 above.
Line 300: Compare Fig. 4P with 4I (and also seen in the Arachnoxylon specimen of Fig 4K). Note the difference in that a permanent strand remains in each rib of Asteropteris and Arachnoxylon with paired protoxylem strands ultimately becoming part of departing traces. A similar pattern is not apparent in Denlongia, although I haven’t studied the latter to know if this difference is important or not.
Line 316: You might correct me on this, but the Venezuela region might in fact have been part of Laurussia in the Devonian, later separated.

Reviewer 2 ·

Basic reporting

This paper is generally very well structured and clearly written (I have noted a few corrections by line number, indicated in Comments to the Authors). The comparison of Keraphyton with other iridopterid taxa is particularly clear. The figures are excellent, and require no revision.

IMPORTANT CORRECTION: The authors make a very clear case for describing a new genus and species for their specimen. They have followed all the technical requirements for describing a new taxon under the ICN. HOWEVER, their species is orthographically incorrect. Since the species is named in honor of a woman, Ruth Mawson, the epithet should be spelled: mawsoniae, in the feminine genitive singular case (See Art 60.8(b) of the ICN (https://www.iapt-taxon.org/nomen/pages/main/art_60.html) Please correct on lines 21: 102: 109: 195: 258: & elsewhere in the paper.

The only portion of the paper that should be strengthened is in the the discussion of the relationships between the ferns, cladoxyls, iridopterids and sphenopsids (lines 31-43). The discussion as written is rather confusing and light on details. There are several questions at issue here: 1) the relationship between iridopterids and sphenopsids; 2) whether the 'permanent protoxylem group' (Moniloformopses) represents a clade and 3) whether sphenopsids are ferns. The comment "The relevance of the Moniliformopses has been heavily discussed (Pryer et al., 2001; Rothwell & Nixon, 2006)" points to the dispute regarding 2 & 3, but the comment is cryptic, and the references cited are out of date. If the authors want to wade into this debate (although it would add context, it's not strictly necessary for the purposes of this paper), they should cite more up-to-date literature. The attempt of Rothwell & Nixon (2006) to critically evaluate gene trees has been superseded by numerous molecular phylogenies since, virtually all recovering sphenopsids as a basal clade within a broad, monophyletic fern clade (See PPGI: http://dx.doi.org/10.1111/jse.12229 & references within, as well as Qi & al. (2018): https://www.sciencedirect.com/science/article/pii/S1055790318301854 and Shen & al. (2018): https://www.ncbi.nlm.nih.gov/pmc/articles/PMC5795342/).

Perhaps more importantly for the purposes of this paper would be a somewhat expanded discussion of the relationship between iridopterids and sphenopsids. As the authors note, the relationship between the two groups is somewhat tentative, but it remains a viable hypothesis, and it strengthens the significance of iridopterids as not just curious Devonian plants but as progenitors of an important modern group of terrestrial plants. Regardless of whether the 'permanent protoxylem group' is a clade, it clearly unites cladoxylopsids and iridopterids, and a brief discussion of this, especially from the developmental perspective, would also be useful.

Experimental design

This is a descriptive paper of a fossil plant, so experimental design is not applicable. Significance of fossil is clearly defined and described.

Validity of the findings

No comment. The validity and novelty of the new taxon is clearly laid out.

Additional comments

The Middle and Late Devonian was a time of rapid radiation of terrestrial plants, setting the stage for the appearance of modern major groups. In their description of a new genus and species of Iridiopteridales from the Late Devonian of Australia, Champreux et al. make a valuable contribution to our understanding of the early evolution of modern ferns sensu lato, as well as the diversity and paleogeography of Late Devonian megafloras.

The rest of these comments are mostly minor corrections indicated by line number.

22-23: "It also supports the view that this flora was, at least, partly specific to this part of the world at this time." Sentence unclear. Do you mean that Late Devonian Australian floras displayed significant endemism?

41: "The discovery of new Devonian fossils affiliated" WITH

66: Mandowa Mudstone [This is apparently a formal name, so I think that 'Mudstone' should be capitalized.]

103: and φυτό (phyto), plant.

125: Mandowa Mudstone

211: a hormonal [the 'h' is aspirated ('huh') so is preceded by 'a']

279-280: "On the contrary" -> better to phrase this, "in contrast"

284-286: Do the authors want to comment on the functional significance of the broad metaxylem tracheids?

289: COMPRISING [not 'comprised of' - comprise is an active verb meaning the same thing as 'encompass']

315: only COMPRISES Compsocradus ...

318: Mandowa Mudstone

Reviewer 3 ·

Basic reporting

The paper is well written, concise and conforms to the standards of Peer J in terms of clarity, presentation of methods, data and conclusions. The photos and diagrams are of good quality. The authors present an interesting interpretation of this fossil. I think some alternatives should be considered, and at least briefly presented, and that some parts of the interpretation are not as certain as presented. I strongly recommend that the images be more extensively labelled. This is elaborated below.

1. The specimen shows signs of distortion- parts of the xylem ribs are broken, some cells are squashed. The authors present information, including their view of what the stele would look like if not distorted, well. The diagrams are helpful.
2. The division within each rib/arm is unusual and should be described a little more carefully. I am not sure one should use the term “ultimate” on a product of dichotomy; however, other terms I can think of are equally problematical. I recommend in line 150 that another sentence or two be inserted to describe the two ways the dichotomies occur in the main ribs, because it jumps too suddenly into how there are two of each type.
3. The emission of small branches off the fundamental ribs is a unique feature of this plant. It really does not occur in plants usually included in iridopterids.
4. Having a central region is unusual- a limited amount is present in a few iridopterids but not in many. Two plants that are rather similar- and possibly should also be compared- are Dixopodoxylon (Middle Devonian), whose affinities are uncertain and which may represent a root, and the Early Devonian Gothanophyton, probably a basal euphyllophyte whose affinities are uncertain. Lateral traces are known in the latter.
5. This leads me to ask, have the authors considered that the structure might be a root, especially given the presence of an endodermis. The authors should state why they don’t think it is a root.
6. I also question if this is an iridopterid- it lacks any evidence of producing laterals and how laterals occur anatomically or morphologically is important in defining iridopterids. It anatomy is rather different too. Might it also represent a unique lineage (if it is a stem).
7. In line 196, the authors suggest emission of laterals is opposite and decussate but there is very little evidence in favor of that. It depends on which ribs produce traces at which level, and whether one or several traces supplied laterals. One could argue equally for helical or whorled as well.
8. One more small question- line 153: cells are xx wide?
9. Concerning labels: label all ribs, not just some of them in Fig. 1 and 2. For example, lateral 1b is not labelled in Fig 1B. is one of the laterals (1D) printed in reverse? Add some arrows to point out specific structures.

Experimental design

OK

Validity of the findings

See item one above; back off on interpretation by stating "possible" but include alternatives.

---

## Round 0.2 · Minor Revisions

Thank you for your careful revisons. We are fortunate to have received a second review from the original 3rd reviewer, who reports that only minor revisions are required. Please attend to these items, and no further peer reviews should be needed. In addition, the labels on the plates are still rather indistinct, e.g., Fig. 3E-H. Simple black text on white background, or the inverse, would be far easier to read than the semi-transparent label backgrounds. Another technique is to use black strokes or 'glows' on white letters (or the inverse).

As always, please take this opportunity once again to 'copy-edit' and carefully check the paper for any minor writing errors, keeping in mind that PeerJ does not provide copy-editing.

I look forward to your revisions and wish you and your families good health at this time of tremendous uncertainty worldwide.

Best regards,

Peter Wilf

Reviewer 3 ·

Basic reporting

This is a second review. The authors have addressed many of the reviewers' concerns. I still have one statement which they don't have to address and a few questions they could-
statement: in reference to fern-like, while it depends on what the authors mean, if related to ferns then I think any relationship to ferns is by default and I am in a minority of not thinking all so-called moniloforms are monophyletic and that they relate to ferns. I think we may find that many represent individual euphyllophyte lineages that began and ended...this plant might be one of them. It may very well turn out to be a member of an as yet unrecognized lineage.

questions/comments:
line 141is this Qi or QIU???
line 387: are the higher beds in the formation similar in lithology? is deposition continuous (or can that be determined). For stating late Late Devonian it seems it could be as old as Frasnian and no younger than LL Devonian or Tn1a, but the data are sparse
line 387, 388: after suggesting (or so I thought) POSSIBLE iridopteridalean, language here is more "certain iridopterid"... I suggest inserting "possible" before iridopteridalean, and in line 388 "may indicate...." because there are some differences.
line 391- again perhaps use "Iridopterid-like plant"?

the opposite, decussate nature is one way the actinostele can be viewed, and this is rather supported by the presence of a central segment- but the array of arms and their divisions also would argue for a more helical departure of laterals. So I think that aspect should be clearly indicated as the authors' interpretation.

Thus I recommend accept after minor revision.

Experimental design

OK

Validity of the findings

OK with the above caveats regarding interpretation of data, although authors support their interpretations

---

## Round 0.3 · accepted · Accept

Thank you for your careful attention to the recent reviewer comments, for the helpful new figure (Fig. 5), and for making the figure labels much more clear. Please note that there is a missing conjunction here and add it in on the next (production) round: "In Fig. 5B, two different types of lateral organs are produced per node, a small type showing the two traces generated by the short branches AND a large type showing the traces generated by the long branches."

Congratulations on this very nice paper, I look forward to seeing it published!

Peter Wilf